# Abnormal and Changing Information Interaction in Adults with Attention-Deficit/Hyperactivity Disorder Based on Network Motifs

**DOI:** 10.3390/brainsci13091331

**Published:** 2023-09-15

**Authors:** Xubin Wu, Yuxiang Guo, Jiayue Xue, Yanqing Dong, Yumeng Sun, Bin Wang, Jie Xiang, Yi Liu

**Affiliations:** 1College of Computer Science and Technology, Taiyuan University of Technology, Taiyuan 030024, China; wuxubin0066@link.tyut.edu.cn (X.W.); xuejiayue0062@link.tyut.edu.cn (J.X.); dongyanqing0078@link.tyut.edu.cn (Y.D.); sunyumeng0466@link.tyut.edu.cn (Y.S.); wangbin01@tyut.edu.cn (B.W.); 2School of Software, Taiyuan University of Technology, Taiyuan 030024, China; guoyuxiang@tyut.edu.cn; 3Department of Anesthesiology, Shanxi Province Cancer Hospital, Taiyuan 030013, China

**Keywords:** ADHD, brain effective network, network motifs, information interaction, node roles, abnormal interaction, changing roles

## Abstract

Network motif analysis approaches provide insights into the complexity of the brain’s functional network. In recent years, attention-deficit/hyperactivity disorder (ADHD) has been reported to result in abnormal information interactions in macro- and micro-scale functional networks. However, most existing studies remain limited due to potentially ignoring meso-scale topology information. To address this gap, we aimed to investigate functional motif patterns in ADHD to unravel the underlying information flow and analyze motif-based node roles to characterize the different information interaction methods for identifying the abnormal and changing lesion sites of ADHD. The results showed that the interaction functions of the right hippocampus and the right amygdala were significantly increased, which could lead patients to develop mood disorders. The information interaction of the bilateral thalamus changed, influencing and modifying behavioral results. Notably, the capability of receiving information in the left inferior temporal and the right lingual gyrus decreased, which may cause difficulties for patients in processing visual information in a timely manner, resulting in inattention. This study revealed abnormal and changing information interactions based on network motifs, providing important evidence for understanding information interactions at the meso-scale level in ADHD patients.

## 1. Introduction

Attention-deficit/hyperactivity disorder (ADHD) is one of the most common neurodevelopmental disorders [1,2,3,4]. The psychopathology of this disorder is marked by developmentally inappropriate and pervasive expressions of inattention, overactivity, and impulsiveness, which result in lower academic achievement and a range of social dysfunctions [5,6,7]. At present, the pathogenesis has not been elucidated.

The inclusion of noninvasive neuroimaging techniques and graph-theoretical analytical methods has enabled the widespread use of large-scale functional brain networks in the research of ADHD [8,9,10,11,12,13,14]. Some studies have proven that ADHD patients have abnormal brain network topology in functional networks, which increases modularity, decreases global efficiency, and increases local efficiency in comparison to normal control (NC) [15,16,17,18,19]. Comparing this to undirected functional networks, direction increases our understanding of the association between brain function and behavior [20]. Some studies have confirmed the abnormal effective connectivity of certain brain regions and the directional causal relationships among ROIs in ADHD patients, which could differentiate patients with ADHD from NC [21,22,23,24]. However, the majority of these studies have described macro- and micro-scale network properties in functional networks in these individuals, and the underlying meso-scale characteristics of brain function networks in ADHD patients are not well known.

Network motifs are recurring local connectivity patterns in a network that are present in numbers significantly higher than those in random networks. They are thought to be the building blocks of brain networks, with functions representing communication channels between multiple nodes [25,26]. In the human functional brain network, Friedman et al. presented the first application of directed network motifs and showed that frequencies of specific directed network motifs could be used to distinguish between patients with Alzheimer’s disease (AD) and NC [27]. Moreover, Wei et al. found the distribution in distinct patterns of several significantly recurring motifs within the network, which supports intra- and inter-module functional connectivity, promoting integration and segregation in network organization [28]. Nonetheless, the above studies only confirmed the frequencies of motifs and analyzed the distribution in the whole brain network [27,28,29,30,31]. The node functions involved in the network motifs of ADHD patients, which could reflect specific brain regions’ information interaction patterns, require further exploration. In this study, the term “information interaction” refers to the process of receiving and transmitting information between brain regions.

To address this gap, we aimed to investigate functional motif patterns in ADHD to unravel the underlying information flow and further analyze motif-based node functions to identify the lesion site of ADHD. In this paper, we focused on (i) identifying functional motif patterns to determine underlying information flow in directed functional brain networks; (ii) introducing the concept of node roles to describe the information interaction functions and mine differential brain regions; (iii) exploring the discrepant effective connectivity based on node roles between specific brain regions; and (iv) probing the node role variation tendency of specific brain regions between ADHD and NC. We investigated abnormal and changing information interactions based on node roles, hoping to provide new insights into the pathogenesis of ADHD.

## 2. Materials and Methods

### 2.1. Participants

We examined resting-state functional MR images (fMRI) using two datasets to verify the consistency of the network motifs in the human brain. (1) The first dataset was the HCP retest dataset, which contains 45 resting-state functional MRI (rs-fMRI) scans from the Human Connectome Project (HCP) Retest data release [32]. They could be identified twice to reflect the consistent motif patterns of the human brain and avoid randomness. Details of the dataset can be found on the HCP website (http://www.humanconnectome.org/, accessed on 30 July 2023). (2) The second dataset consisted of 50 rs-fMRI scans from NC and 42 rs-fMRI scans from individuals with ADHD. It was compared to the motif detection results of HCP datasets in order to establish a foundation for further research. They were provided by the Consortium for Neuropsychiatric Phenomics study at the University of California, Los Angeles (UCLA). Details of the dataset can be acquired from the OpenfMRI data-sharing webpage (https://www.openfmri.org/, accessed on 30 July 2023).

When the respective researchers released the data, they stated that all participants gave written informed consent after receiving a thorough explanation. Specific demographic characteristics of the two datasets are shown in Table 1.

### 2.2. Data Acquisition and Preprocessing

The HCP imaging data were acquired on a customized 3T Siemens connectome-Skyra 3T scanner using a multiband sequence (Berlin, Germany). Each participant completed two rs-fMRI scanning sessions 140 days apart. The parameters were as follows: repetition time (TR) = 720 ms, echo time (TE) = 33.1 ms, slice thickness = 2 mm, slices = 72, flip angle = 52°, and duration = 14 min and 33 s (1200 TRs). The magnetic resonance images of ADHD patients were acquired using a 3T Siemens Trio scanner (Berlin, Germany). Each participant completed one rs-fMRI scanning session. The parameters were as follows: repetition time (TR) = 2 s, echo time (TE) = 30 ms, slice thickness = 4 mm, slices = 34, flip angle = 90° field of view (FOV) = 192 mm, matrix = 64 × 64, and duration = 5 min and 4 s (152 TRs). During the entire scanning process, the participants of the two datasets were asked to relax and close their eyes, but not fall asleep.

Data preprocessing of the two datasets was conducted using the DPABI toolbox (Data Processing & Analysis for Brain Imaging, http://rfmri.org/dpabi, accessed on 30 July 2023) [33]. Firstly, we discarded the first ten volumes of the signal, considering the adaptability of the environment, and corrected the first slice timing and head motion of the remaining data. Subsequently, the data generated were normalized relative to the Montreal Neurological Institute (MNI) standard space in order to compensate for individual brain variations. Next, to reduce the inexactness of registration and enhance the signal-to-noise ratio, spatial smoothing was conducted using a Gaussian kernel with 6 mm full-width at half-maximum (FWHM). Additionally, bandpass filtering (0.01 ≤ f ≤ 0.1 Hz) was implemented on the image to transform the time series into the frequency domain and calculate the energy in the lower frequency band. Finally, resting-state scans were parcellated into 90 regions of interest (ROIs) using the automatic anatomical marker template (AAL atlas) [34] and the time series were extracted. Notably, the cerebellum was excluded from this study.

### 2.3. Construction of Directed Brain Functional Networks

To construct directed functional brain networks using resting-state fMRI data, we used the Wiener–Granger causality analysis (GCA) approach to estimate the mean time series of any pair of ROIs by computing the causation, which is a common method to study the causal relationship between variables on the base of the causality of time series. Compared to the traditional causality algorithms (e.g., structural equation modeling and dynamic causal modeling), GCA does not necessitate the postulation of an effect between any two regions [35]. It is flexible and could accommodate hemodynamic variability [36]. The results of causal relationships are reflected in the form of predictions. Here, GCA was employed for every ROI in the whole brain, and the effective connectivity (EC) from each ROI to the remaining ROIs was obtained. The GCA of 2 time-series $X_{t}$ and $Y_{t}$ can be calculated using the following autoregressive model:(1)$$Y_{t}=\overset{p}{\underset{i=1}{\sum }}\;A_{11,i}X_{\left(t-i\right)}+\overset{p}{\underset{i=1}{\sum }}\;A_{12,i}Y_{\left(t-i\right)}+\varepsilon_{t}$$
(2)$$X_{t}=\overset{p}{\underset{i=1}{\sum }}\;A_{21,i}Y_{\left(t-i\right)}+\overset{p}{\underset{i=1}{\sum }}\;A_{22,i}X_{\left(t-i\right)}+\varepsilon_{t}^{\prime }$$
where *p* is the model order of the maximum number of lagged samples; $A_{11,i}$ and $A_{21,i}$ are the signed-path coefficients; $A_{12,i}$ and $A_{22,i}$ are autoregression coefficients; and $\varepsilon_{t}$ and $\varepsilon_{t}^{\prime }$ are residuals for each time series. If the variance in $\varepsilon_{t}$ (or $\varepsilon_{t}^{\prime }$) is reduced by the inclusion of the Y (or X) term in the first (second) equation, then we could conclude that Y (or X) G causes X (or Y). Assuming that X and Y are covariance stationarity, the magnitude of the interaction can be determined by calculating the log ratio of the prediction error variances for the restricted (R) and unrestricted (U) models [37]:(3)$$F_{X\to Y}=\ln\frac{\mathrm{var}\left(\varepsilon_{R}\right)}{\mathrm{var}\left(\varepsilon_{U}\right)}$$
(4)$$F_{Y\to X}=\ln\frac{\mathrm{var}\left({\varepsilon \prime }_{R}\right)}{\mathrm{var}\left({\varepsilon \prime }_{U}\right)}$$
where $\varepsilon_{R}$ and ${\varepsilon \prime }_{R}$ are derived from the model omitting the $A_{11,i}$ and $A_{21,i}$ (for all i) coefficients, and $\varepsilon_{U}$ and ${\varepsilon \prime }_{U}$ are derived from the full model. It is worth noting that the results of constructing a directed brain functional network using GCA may be better if the fMRI signals are first deconvolved with the hemodynamic response. Additionally, the selection of model order p is very important. If the order p of the multivariate autoregressive model is too small, the data may not be accurately represented, and if it is too large, the model estimation could be problematic. In this study, the model order p was set to 1 and the calculation process is detailed in the Appendix A.

To make the strength and reliability of the general results certain, the coefficient matrix of the directed brain functional networks was converted into binary form with varied thresholds. The equal interval threshold range (with 10% to 40% as limits and a gap of 5%) was taken into consideration for calculation. Notably, the brain functional networks in different groups were typically formed with the same threshold to make sure that all produced networks had similar topological structures with the same amount of edges [38].

### 2.4. Identification of Network Motifs

Complex networks are studied across many fields of science. To uncover their structural design principles, “network motifs” were defined; they are patterns of interconnections occurring in complex networks at numbers that are significantly higher than those in randomized networks [25]. Such motifs have been found in networks in the fields of biochemistry, neurobiology, ecology, and engineering. As a kind of complex network, we will further investigate the motif patterns in directed human functional brain networks.

Within a network, a motif is a small graph which is local to the network and consists of M nodes as well as a collection of edges linking them [26]. Keeping to a motif size of M, the amount of motif classes remains unchanged. Herein, our main focus is on the 13 different 3-motifs, as shown in Figure 1A. For comparison with previous studies, the IDs of the motifs are consistent.

Initially, we figured out the motif frequency distribution by tallying the number of times each type of three-node motif appeared in the network. After that, we compared the frequency distribution of this result with those from 1000 surrogate random networks. The random networks serve as a null model to determine which motifs are overexpressed in the original network, which retains an identical number of nodes and edges, along with the same in-degree and out-degree distributions [25,28]. Given the motif frequency in the random networks, the magnitude of overexpression of motif M in G is given by its z-score:(5)$$Z_{M}=\frac{N_{M}^{\mathrm{real}\;}-\left\langle N_{M}^{\mathrm{rand}\;}\right\rangle }{\mathrm{std}\left(N_{M}^{\mathrm{rand}\;}\right)}$$
(6)$$\left\langle N_{M}^{\mathrm{rand}\;}\right\rangle =\frac{1}{1000}\overset{1000}{\underset{1}{\sum }}N_{M}^{\mathrm{rand}\;}$$
(7)$$\mathrm{std}\left(N_{M}^{\mathrm{rand}\;}\right)=\sqrt{\frac{\sum_{1}^{1000}\left(N_{M}^{\mathrm{rand}\;}-\left\langle N_{M}^{\mathrm{rand}\;}\right\rangle \right)}{999}}$$
where $N_{M}^{\mathrm{real}\;}$ is the occurrence frequency of motif M in the real network, and $\left\langle N_{M}^{\mathrm{rand}\;}\right\rangle$ and $\mathrm{std}\left(N_{M}^{\mathrm{rand}\;}\right)$ indicate the mean and SD of its frequency of appearance in the 1000 surrogate random networks. Motif M is regarded as statistically significant only if Z_M_ > 1.96 (*p* < 0.05) [28].

To verify the consistency of the motifs, we first performed motif identification twice in the directed functional brain networks of the HCP retest dataset to identify the specific motif patterns in the human brain, reflecting the underlying information flow. In this study, the coefficient matrix of the HCP-directed brain functional networks was converted into binary form by applying a 30% threshold to ensure that the number of edges in the brain network matched that of the ADHD dataset. For clarity, the 30% threshold is in reference with keeping 30% of the strongest connection. Then, the same procedure was performed on the ADHD dataset, and the motif detection results for both datasets were consistent. This indicates that the recognized motifs are indeed specific organizational patterns in the directed brain functional network. Moreover, we identified motif patterns in all networks of ADHD groups (threshold ranging from 10% to 40% with a partition interval of 5%) and found that when the threshold was 30%, the population proportion of all kinds of network motifs in the two groups reached the maximum and remained stable. The motifs of other threshold networks in the brain atlas are shown in Appendix A.

### 2.5. Analysis of Motif-Based Node Roles

In order to better characterize the node functions participating in network motifs, we introduced the concept of node roles based on the similarity of ties between participating nodes. In other words, different node roles represent different ways in which information interacts. In this study, according to the results of identifiable motif patterns, we set the node roles into three types, which were named “*Double Ping-Pong*” (DPP), “Ping-Pong_Output” (PPO), and “Ping-Pong_Receive” (PPR) (Figure 1B), and calculated the Pearson correlation with the traditional node degree to describe the functions of the node roles (see Appendix A). As shown in Figure 1B, the “*Double Ping-Pong*” role, the first type of node role, has two bidirectional edges that can receive and output information from the other two nodes. The “Ping-Pong_Output” role, the second type of node role, has one bidirectional edge and one outward edge, and is more inclined to output node information. The “Ping-Pong_Receive” role, the third type of node role, has one two-way edge and one inward edge, which is more inclined to receive node information. The measurement of motif-based node roles is obtained from their role-degree, which is the frequency of node roles appearing in the network:(8)$${RD}_{i}=\overset{n}{\underset{j=1}{\sum }}\;\overset{m}{\underset{p=1}{\sum }}\;{M\;}_{i,jp}$$
(9)$${M\;}_{i}=\;{f\;}_{i}$$
where i is the type of node role, j represents the category of identifiable motif, p represents the number of motifs in the brain network, and f indicates the number of node roles in a motif. The M indicates that this motif contains a number of corresponding node roles. For example, motif13 contains three DPP roles. The role-degree indicates the information interaction capability. The greater the role-degree, the stronger the interaction capability.

Furthermore, in order to better explore abnormal information interactions between specific brain regions, we analyzed the participating edges of node roles (Figure 1B). The measurement is obtained from their edge-degree, which is the number of times that directed edges appear in the node role:(10)$${ED}_{i}=\overset{k}{\underset{s,t=1,s\neq t}{\sum }}(ua_{i,st}+ba_{i,st\;})\;$$
where i is the type of node role, s represents the source node, t represents the target node, u indicates the presence of a unidirectional edge between nodes, and b indicates the presence of a bidirectional edge. $a_{st}$ indicates the number of connections from the source node to the target node. When calculating the bidirectional edges between nodes, the value of b is set to 1 and u to 0, and vice versa. The greater the value of the number, the more frequent the information interaction between specific brain regions.

In addition, we researched the variation tendency of node roles to explore the changing interaction ways of specific brain regions. We calculated the weights of different roles at each node and determined which role dominates the information function of nodes through a one-sample *t*-test. The formula for determining the dominant role of each node is as follows:(11)$$RW_{i}=\frac{RD_{i}}{{\sum_{i=1}^{3}\;RD}_{i}}$$
(12)$$t=\frac{\left\langle RW\right\rangle -\mu_{0}}{\mathrm{std}\left(RW\right)/\sqrt{n}}$$
where i is the type of node role, and $RW_{i}$ indicates the weight of the node role in each node. $\left\langle RW\right\rangle$ and $\mathrm{std}\left(RW\right)$ denote the average value and the SD of the node role’s weight in each group, $\mu_{0}$ represents the assumed population mean (set to 0.33), and n represents the number of samples. We convert the t value to the corresponding *p* value. When *p* is less than 0.05, it indicates that the weight of the node role is significantly higher than $\mu_{0}$ and we consider it to be the dominant role of the node.

By calculating the role weights of each node in a directed functional network, we could determine the relative contribution of different roles to the information processing of each node. This could be used to better understand the function of the brain and to identify potential targets for therapeutic interventions in disorders. BrainNet Viewer was employed to complete the visualization [39].

### 2.6. Statistical Analysis between Groups

In this study, we used the Wilcox test to evaluate differences between groups. Since gender is a categorical class of data that is not ordered, chi-square distribution was used to test for differences between groups. Spearman’s correlation, a statistical method that does not rely on specific assumptions about the data, was used to assess the relationship between cognitive factors. All results were adjusted for multiple comparisons using the Benjamini and Hochberg false discovery rate (BH_FDR) approach. The threshold for determining significant differences was set at *p* < 0.05. Please see Appendix A for more detail. The pipeline of the analysis strategy for this study is displayed in Figure 2.

## 3. Results

### 3.1. Motif Patterns within Directed Functional Brain Networks

We examined the distribution patterns of 13 classes of three-node motifs in the HCP retest dataset and the ADHD dataset. The frequency distribution of the three-node motifs of the directed functional brain networks in the two datasets is shown in Figure 3A. Compared with the matched random networks (Figure 3B), five motifs (ID = 4, 6, 9, 12, and 13) were observed at frequencies significantly higher than expected (Z > 1.96, *p* < 0.05) (Figure 3C), which reflects the underlying information flow. The proportion was obtained by dividing the number of subjects with a Z-score greater than 1.96 by the total number of people. The results of the identified motif patterns in the two datasets were consistent, as shown in Figure 3D.

It is essential to keep in mind that a motif that has no significance cannot be considered a building block of brain networks. In Figure 3, it can be seen that motif IDs = 1, 2, and 3 are also more numerous than motif IDs = 12 and 13. Although they occur frequently, they occur with a similar frequency during the process of rewiring through random links.

### 3.2. Differential Brain Regions about Node Roles between Groups

To better understand the information interaction capability of brain regions between the ADHD patients and the NC, we analyzed the role-degree data and noticed some brain areas which had been impacted by the disease (Figure 4). When controlling for multiple comparisons (FDR), four regions were observed to be significantly different in terms of the DPP and PPR roles, and two regions were significantly different in regard to the PPO role. Differentiated areas of the brain in the DPP role included the left precentral gyrus, right hippocampus, right amygdala, and left precuneus (Figure 4A). For the PPO role, the significantly different brain regions included the left Rolandic operculum and the left Heschl’s gyrus (Figure 4B). For the PPR role, notable variations in brain areas included the left inferior frontal gyrus, triangular part, right lingual gyrus, right caudate nucleus, and left inferior temporal gyrus (Figure 4C). The distribution of differential brain regions in each RSN is shown in Figure 4D. Please see Appendix A for more detail.

### 3.3. Discrepant Effective Connectivity of Node Roles between Groups

In order to explore the abnormal effective connectivity of information interaction between specific brain regions, we compared the edge-degree data participating in node roles and found that there were significant differences in the connections between brain regions in the two groups (Figure 5). Connections between the brains with significant differences for the DPP role included ten pairs of bidirectional connectivity. In particular, compared to the NC, two brain regions between the left precentral gyrus and the left precuneus interacted less, and interaction frequencies between the right hippocampus and the right amygdala increased. For the PPO and PPR roles, we found differences in one-way connections between four pairs of brain regions, respectively. Specifically, for the PPO role, the left Rolandic operculum output less information to the right median cingulate and paracingulate gyri in ADHD patients. For the PPR role, the left inferior frontal gyrus triangular part received an increased frequency of information from the left amygdala compared to NC. Please see Appendix A for more detail.

### 3.4. Variation Tendency of Node Roles between Groups

Each node has three types of roles, and diverse roles characterize different information interaction methods. To explore the relative contribution of diverse roles to the information processing of each node, we analyzed nodal role weight and found that there were certain variation tendencies in some brain regions (Figure 6). Specifically, the bilateral thalamus, which plays a DPP role in normal subjects, takes on a PPO role in ADHD patients. Additionally, the right middle occipital gyrus, left supramarginal gyrus, and left inferior temporal gyrus shifted from a DPP role in NC to a PPR role in ADHD patients. These changes in brain function may contribute to the symptoms associated with ADHD. Please see Appendix A for more detail.

### 3.5. Correlations between Node Role-Degree and ASRS Scores

Although the above results indicated that the information interactions within the brain’s functional network in ADHD patients were abnormal, we were still curious about the connections between node role-degree and cognitive impairment (measured by ASRS) in ADHD patients (see Figure 7). We primarily focused on areas that showed a statistical difference between the two groups. Based on the Spearman correlations, in regions where there were significant differences, the ASRS scores showed significant positive correlations with two nodes in relation to the DPP role-degree. One node had strong positive correlations, while another node had strong negative correlations in terms of the PPR role-degree. Please see Appendix A for more detail.

## 4. Discussion

In this study, we examined motif patterns in the directed functional brain networks in order to uncover meaningful underlying information flows of brain activity, and mined divergent and varied information interactions based on node roles between groups. Several functional motifs were determined to be statistically significant within the network. Then, based on the identification results of the network motifs, we introduced the role concept to characterize the information interaction functions of nodes and to explore the diseased brain regions in the information processing of the patients. As opposed to the classical effective connectivity, the node roles demonstrate the information interaction between multiple nodes (in this study, three nodes in particular). It denotes interconnectivity among sets of nodes, not just between two nodes, hoping to provide new insights into the pathogenesis of ADHD. There may be potential for using this information to develop targeted interventions for ADHD.

### 4.1. Significant Motif Patterns in the Directed Functional Brain Networks

Upon inspection of the directed functional brain networks, we noticed that the occurrence of five types of three-node motifs was significantly more likely to be embedded in the functional brain networks of the two datasets compared with random networks, and the results of the identified network motifs were consistent [28]. This indicates that these five types of motifs are the basic building blocks of directed human functional brain networks, which play important functions in information processing and regulation. These five motifs can be split into chain motifs and loop motifs. Investigations conducted previously have indicated that the three categories of chain-like motifs largely contribute to the unification of information within the whole-brain network. Additionally, loop-like motifs may be the critical information-processing pattern that enables the local integration of functionally related regions, leading to greater functional detail and a more enriched functional state [26,28,40,41,42,43,44]. This has a notable significance in uncovering the information interaction principles of brain functional networks.

### 4.2. Differential Brain Regions Regarding Information Interaction Capability in ADHD Patients

Different node roles represent different information interaction ways. The node role-degree indicates the information interaction capability; we found that some brain regions of ADHD patients had significant differences compared with those of NC subjects. For the DPP role, the left precentral gyrus and the left precuneus in the patients were significantly decreased. These are the components of the default network, which is mainly responsible for emotional processing and self-introspection [45]. When the default network is damaged, it may cause poor self-control and be accompanied by emotional problems. However, the interaction functions of the right hippocampus and the right amygdala were significantly increased. We hypothesize that the functional abnormalities may be caused by a reduced volume in the hippocampus and amygdala [46,47,48,49,50]. Additionally, both of these brain regions are in the core of the limbic system, which is responsible for controlling emotions. The anomaly to the limbic system could lead patients to develop mood disorders. In addition, the amygdala is also a center for emotional processing and is involved in the production and expression of emotions [51,52]. Some studies have found that ADHD patients have difficulty recognizing fearful facial expressions and have excessive amygdala activity [52]. Figure 7 demonstrates a strong positive relationship between DPP role-degree and ASRS scores, which suggests that the heightened activities in the right hippocampus and right amygdala are likely to be the cause of the symptoms experienced by ADHD patients. For the PPR role’s function with stronger information-receiving ability, the left inferior frontal gyrus triangular part was significantly increased, and the right lingual gyrus, the right caudate nucleus, and the left inferior temporal gyrus were significant decreased. Specifically, the inferior frontal gyrus triangular part is involved in language production [53,54,55]. The brain region received information unusually frequently, which may be one of the reasons for patients’ irritable impulses to speak. Additionally, the strong positive relationship between PPR role-degree and ASRS scores, observed in Figure 7, suggests that heightened interaction leads to the symptoms experienced by ADHD patients. Notably, the right lingual gyrus, which belongs to the occipital lobe, is mainly responsible for visual processing [56,57,58]. The left inferior temporal gyrus plays a role in processing visual information. It receives input from the occipital lobe, which is a higher-order area for visual processing [59,60,61]. The capability of receiving information in two brain regions was decreased, which may cause patients to process visual information in a timely manner, resulting in inattention. Moreover, the data presented in Figure 7 demonstrate a very negative correlation between PPR role-degree and ASRS scores, which strongly suggests that the reduced interaction in the right lingual gyrus is the cause of the symptoms experienced by ADHD patients. The right caudate nucleus is an important component of the striatum. The striatum is a major source of dopamine, which plays a vital role in attention [62,63]. Current neurobiological studies suggest that distraction and restlessness in ADHD patients are associated with decreased dopamine [64]. We hypothesize that the abnormal function of the right caudate nucleus may be related to the decreased volume of the caudate nucleus in patients [46,47]. For the PPO role with a stronger information-outputting function, the left Rolandic operculum and the left Heschl’s gyrus were significantly decreased in patients. They are part of the sensorimotor network, which is primarily responsible for monitoring and regulating motor behavior [65,66]. The weakening of sensorimotor network output information may cause an abnormal reception of information in other brain areas in patients, causing hyperactivity symptoms.

To sum up, the node roles illustrate the interaction of information between the three nodes, which play important roles in the brain’s activity. Through statistical analysis, we found that some brain regions had differential information interaction capability in the process of information processing, resulting in ADHD patients with symptoms such as language impulses, emotional disorders, visual disorders, and attention deficits. The majority of the brain regions with abnormal interactions were also areas of damage in the patients’ brains, which corresponds with results from prior studies. Notably, visual impairments are rarely reported, and patients have abnormalities in processing visual information, which may be the main cause of inattention. Furthermore, utilizing the proposed method on activity tasks as a classic Go-no-Go paradigm could be a way to validate the brain regions that have been largely studied.

### 4.3. Discrepant Effective Connectivity of Node Roles in ADHD Patients

Brain functional networks do not operate in isolation, and there are interactions between brain networks that play an important role in maintaining healthy mental states and cognitive abilities. Based on the node roles, we found that there are some discrepant effective connections during information interaction [21,52,67]. Specifically, in terms of the DPP role, the interaction between the default mode network and other functional subnetworks in ADHD patients decreased. However, the interaction within the subcortical network increased. For the PPR role, the interaction was disordered between the attention network and other functional subnetworks in ADHD patients. For the PPR role, the interaction from the sensorimotor network to the subcortical network decreased in ADHD patients compared to NC. Abnormal interactions intra- and inter-RSN may cause symptoms such as difficulty concentrating, hyperactivity, and impulsive behavior in ADHD patients. In particular, the interaction between the left precentral gyrus and the left precuneus decreased. These two brain regions are responsible for two important areas of executive control function, including inhibiting impulses and regulating attention [68,69]. The effective connection between these two brain regions was impaired, causing them to have difficulty controlling their behavior and attention, while the interaction between the right hippocampus and the right amygdala was increased. Previous research suggests that emotional reactivity is one of the most disabling symptoms associated with attention-deficit/hyperactivity disorder (ADHD) [70,71]. The over-interactions between the hippocampus and amygdala may lead to emotional instability in patients [52]. Moreover, compared with NC, the right caudate nucleus, which receives information from the right inferior frontal gyrus triangular part, decreased, while the left inferior frontal gyrus triangular part, which receives information from the left amygdala, increased. The right caudate nucleus and the left amygdala are related to emotion, reward, and attention, and the triangular inferior frontal gyrus is associated with language production. The above abnormal connections may reflect the brain’s asymmetry in emotional regulation and attention control. This asymmetry may be associated with neurodevelopmental disorders such as ADHD. Notably, the discrepant effective connections, mostly belonging to the frontal striatum and frontal parietal pathways [72,73,74], could lead to impaired cognitive and attention functions. Generally speaking, we could accurately pinpoint which two brain regions have abnormal information interactions through connections based on node roles. It is noteworthy that this is distinct from the classic effective connection. In this research, node role is the basis of the differential effective connection, symbolizing the intercommunication between the three nodes. This provides a new mentality for us to accurately understand the mechanisms of ADHD.

### 4.4. Changed Information Interaction Ways of Brain Regions in ADHD Patients

The node roles based on motifs play their respective functions during the information processing process. We found that some brain regions in ADHD patients have changed in the way information interacts. Specifically, the function of the bilateral thalamus to receive information was decreased, which may be caused by changes in the volume of the thalamus in ADHD patients [23,75]. The thalamus is a key subcortical structure of the cortico-stria-to-thalamo-cortical loop (CSTC) [76], mostly involved in attention and cognitive behavioral processes, and is considered a mediator connecting the basal ganglia, cerebral cortex, and cerebellum [47,77,78]. It regulates brain regions that are responsible for both stimulating and inhibiting functions, thus impacting and altering behavioral outcomes [78,79]. Moreover, the function of the right middle occipital gyrus, the left supramarginal gyrus, and the left inferior temporal gyrus to output information was decreased. Notably, the left inferior temporal gyrus is an important pathological brain area. We found that the PPR role’s function dominates the left inferior temporal gyrus, but, compared with NC, the role’s function was still decreased. This means that the patient’s ability to process visual information may be weakened, which may cause the patient’s attention deficit.

Broadly speaking, according to the distribution of brain regions, there are abnormal brain regions in each RSN except the default mode network. Some studies have also shown that ADHD patients have distributed brain network disorders rather than disorders with discrete region abnormalities [13,80]. This is demonstrated by the alteration of nodes’ roles in specific brain areas. We take it for granted that each node has three separate node roles that serve different purposes in information processing. The patient’s brain region has an abnormal function as a result of changes in the brain area’s information processing. This offers us a novel perspective to accurately grasp the mechanisms of ADHD. Future research may further reveal the information interaction between these brain regions, thus providing a more targeted approach to ADHD treatment.

### 4.5. Limitations and Future Directions

Despite the dependability of some of the outcomes of this study, it still has certain restrictions. First, due to comparisons with previous studies and the complexity of the calculations, this study only focused on three-node motifs. In future research, we plan to expand the motif size to four or five nodes. Then, in order to compare it with the previous literature about motif identification, the cerebellum information was excluded. We will incorporate cerebellar information in further studies. Additionally, despite being at rest, the human brain will alternate between various activities periodically. The latest dynamic network investigations have verified the existence of fluctuations in functional connections, which has gained growing attention in the scholarly world. The analysis of ADHD patients using dynamic methods has yielded certain results that cannot be obtained through static network analysis. For example, Kaboodvand N et al. suggested a dynamical systems approach to examine how the DMN recruits different configurations of network segregation and integration as time passes. Their exploration uncovered that ADHD differs to controls, both in terms of the recruitment rate and topology of specific synergies between resting-state networks [81]. Our further research would focus on the node roles of a dynamic network. Finally, ADHD is typically classified into three categories: inattentive, impulsive/hyperactive, and combined. The various subtypes of ADHD patients have varying pathological mechanisms. For example, Iravani B et al. utilized a freshly created adaptive-frequency-based model of whole-brain oscillations to resting-state fMRI data. The model matched the results of the empirical behavior data, showing two distinct ADHD subgroups with different behavioral phenotypes related to emotional instability (i.e., depression and hypomanic personality traits) [82]. Petrovic P et al. proposed that a top-down dysregulation framework could unify ADHD, emotional features seen in ADHD, and borderline and antisocial personality disorder into one group of mental health issues [83]. When we drill down to the specific subtypes of ADHD, we believe the proposed approach may unearth somewhat different brain regions, or differential connections compared to the results of uncategorized ADHD. We will take this factor into account in the next study.

## 5. Conclusions

We assembled directed functional brain networks and confirmed three-node network motif patterns in two datasets. The results show that motifs with IDs = 4, 6, 9, 12, and 13 are specific interaction patterns in directed human functional brain networks that characterize the underlying information flow within the network. Then, we introduced the concept of node roles based on the similarity of ties in order to better characterize the node functions participating in network motifs. Different node roles represent different information interaction functions. Through statistical tests, we found that some brain regions in ADHD had significant differences compared with NC. The results showed abnormal and changing information interactions in some brain regions, such as the right hippocampus, the right amygdala, the right caudate nucleus, the bilateral thalamus, etc., which may be caused by modifications to the sizes of brain regions. Notably, we also found some discrepant effective connections, mostly belonging to the frontal striatum and frontal parietal pathways involved in attention and executive function. These abnormal interactions can lead to impaired cognitive and attentional functions. This study revealed abnormal and changing information interactions based on network motifs, providing important evidence for understanding information interactions at the meso-scale level in ADHD patients.

## Figures and Tables

**Figure 1 brainsci-13-01331-f001:**
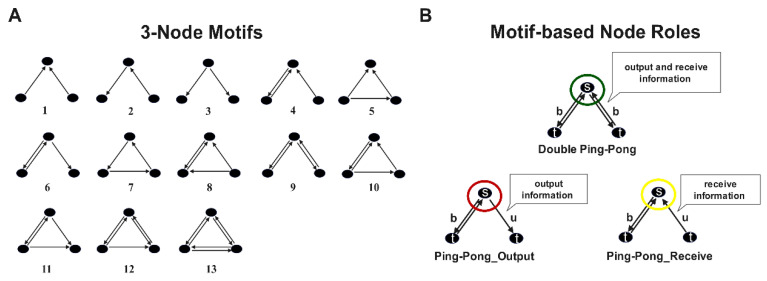
All possible directed 3-node motifs and node roles based on identified motifs. (**A**) Directed 3-node motifs labeled ID 1 to 13. Circles indicate nodes in the network and arrows represent functional relationships between nodes; (**B**) three types of node roles based on identified motif. They are named “Double Ping-Pong”, “Ping-Pong_Output”, and “Ping-Pong_Receive”, where s represents the source node, t represents the target node, b indicates the bidirectional edge, and u indicates the unidirectional edge.

**Figure 2 brainsci-13-01331-f002:**
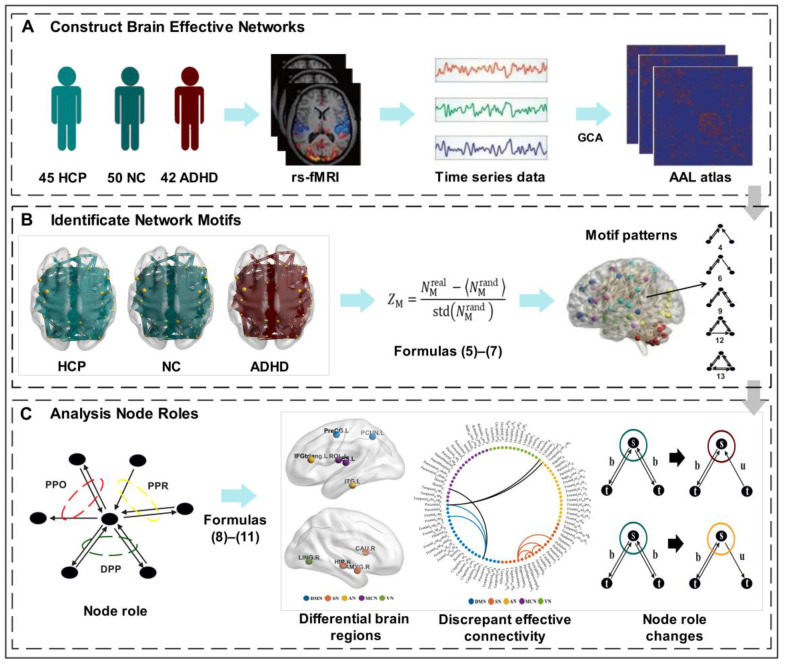
Schematic overview of the analysis strategy. (**A**) Construct brain effective networks by estimating the mean time series of any pair of ROIs using the GCA method; (**B**) identified network motifs of human brain effective networks in two datasets. Through the calculation of Formulas (5)–(7), it was found that the specific network motifs of human brain effective network are 4, 6, 9, 12, and 13; (**C**) analysis of motif-based node roles in ADHD dataset. According to the results of the identifiable motifs, we categorized the node roles into three types: DPP, PPO, and PPR. Through the calculation of Formulas (8)–(11), we used statistical analysis to explore the differential brain regions, discrepant effective connectivity, and changing node roles.

**Figure 3 brainsci-13-01331-f003:**
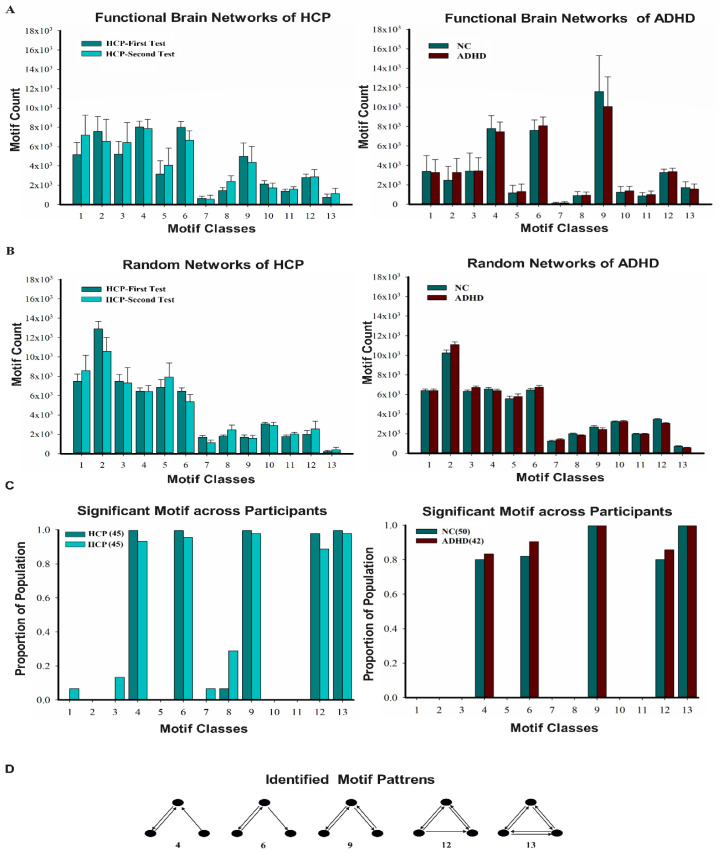
Identification of motif patterns in HCP retest dataset and ADHD dataset. In the HCP retest dataset, the green bar represents the first test data for HCP, while the dark-green bar represents the second test data for HCP. In the ADHD dataset, the dark-green bar represents the data for NC, while the dark-red bar represents the data for individuals with ADHD. (**A**) Frequency distribution (mean ± variance) of 13 classes of three-node motifs; (**B**) frequency distribution (mean ± variance) of three-node motifs in surrogate random networks conserving the same in/out degree distribution; (**C**) proportion of significant motif across participants; (**D**) identified motifs in two datasets.

**Figure 4 brainsci-13-01331-f004:**
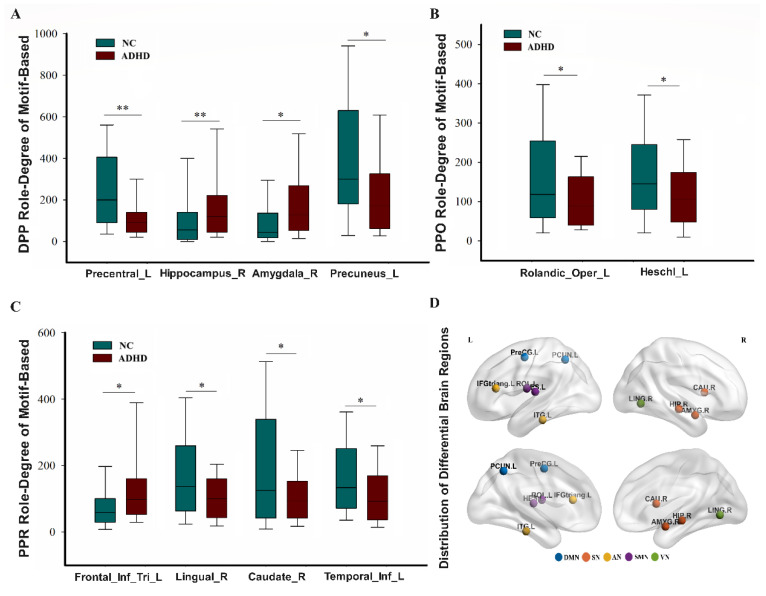
Differential brain regions showing node roles between ADHD patients and NC (*p* < 0.05: *, *p* < 0.01: **) and the distribution in each RSN. (**A**) Differential brain regions of DPP role; (**B**) differential brain regions of PPO role; (**C**) differential brain regions of PPR role; (**D**) the distribution of differential brain regions in the brain map. Different colors represent the 5 RSNs. The 5 RSNs include the DMN, default mode network; SN, subcortical network; AN, attention network; SMN, sensorimotor network; VN, visual network. The PreCG.L, the HIP.R, the AMYG.R, and the PCUN.L belong to the DPP role; the ROL.L and the HES.L belong to the PPO role; the IFGtriang.L, the LING.R, the CAU.R, and the ITG.L belong to the PPR role.

**Figure 5 brainsci-13-01331-f005:**
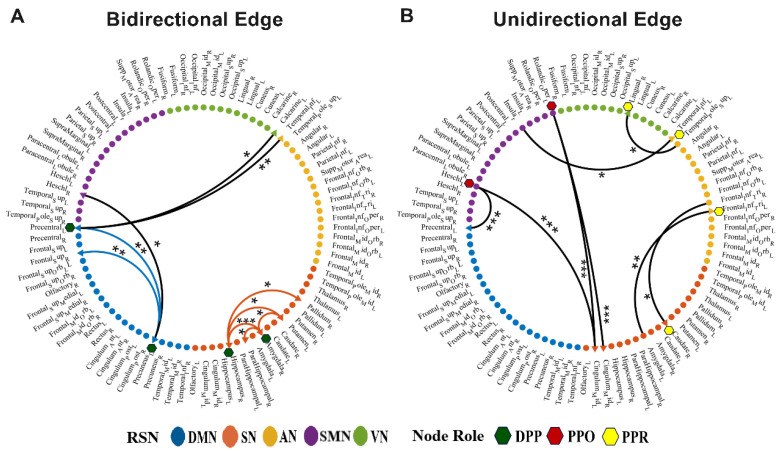
Discrepant effective connectivity between groups (*p* < 0.05: *, *p* < 0.01: **, *p* < 0.001: ***). (**A**) Discrepant bidirectional effective connectivity of the DPP role; (**B**) discrepant unidirectional effective connectivity of the PPO role and the PPR role. Five RSNs are symbolized by different colored dots. The five RSNs include the DMN, default mode network; SN, subcortical network; AN, attention network; SMN, sensorimotor network; VN, visual network. Different colored five-pointed stars represent the three types of node roles. Black lines indicate effective connectivity between RSNs, and lines of other colors indicate effective connectivity within RSNs.

**Figure 6 brainsci-13-01331-f006:**
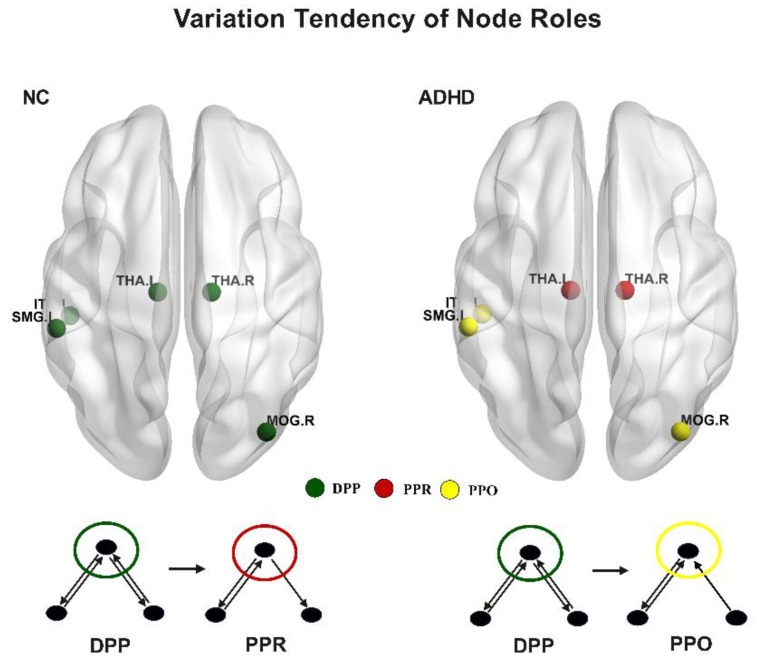
Variation tendency of node roles from NC to ADHD patients. Different colored circles represent the three types of node roles. Green circle represents the DPP role, red circle represents the PPR role, and yellow circle represents the PPO role.

**Figure 7 brainsci-13-01331-f007:**
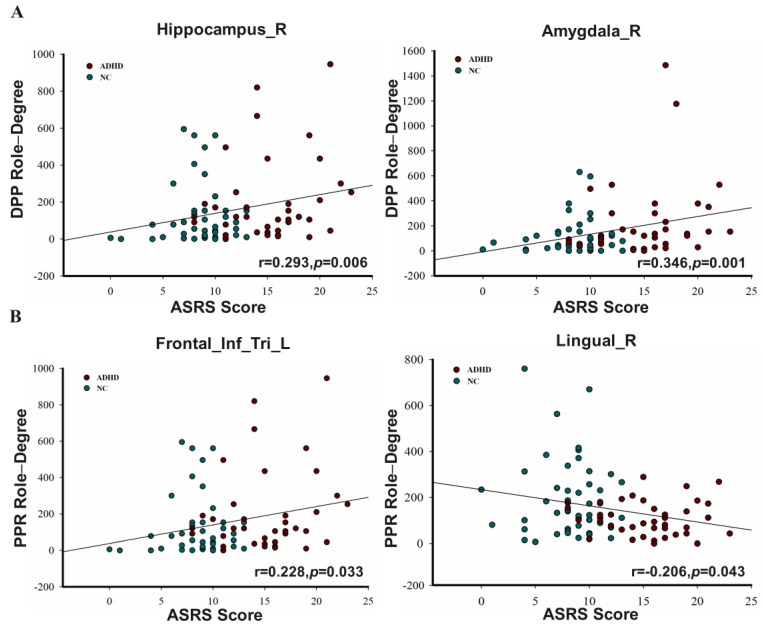
Correlations between node role-degree and ASRS scores. (**A**) For the DPP role, the role-degree of the right hippocampus and the right amygdala have significant positive correlations with ASRS scores; (**B**) For the PPR role, the role-degree of the left inferior frontal gyrus triangular part has significant positive correlations with ASRS scores, while the role-degree of the right lingual gyrus has a strong negative correlation.

**Table 1 brainsci-13-01331-t001:** Specific demographic characteristics of two datasets.

Group	HCP	ADHD	NC	*p*-Value
Number	45	42	50	--
Age (mean ± SD)	30.6 ± 3.17	32.71 ± 10.47	32 ± 8.96	0.728 ^a^
Sex (M/F)	14/31	21/21	27/23	0.702 ^b^
ASRS	--	15.43 ± 3.80	7.94 ± 2.89	0.000 ^a^

Abbreviations: SD, standard deviation; ASRS, Adult ADHD Self-Report Scale. ^a^ Independent-samples *t*-test. ^b^ Pearson chi-square two-tailed test.

## Data Availability

The details of the Human Connectome Project (HCP) Retest data can be found on the HCP website (http://www.humanconnectome.org/, accessed on 30 July 2023). The fMRI data of ADHD were downloaded from the OpenfMRI data-sharing webpage (https://www.openfmri.org/, accessed on 30 July 2023).

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
