# Peer review of "Abnormal and Changing Information Interaction in Adults with Attention-Deficit/Hyperactivity Disorder Based on Network Motifs"

_brainsci, 2023, doi:10.3390/brainsci13091331_

Round 1
Reviewer 1 Report
Wu et al investigated functional motif patterns in directed brain networks of ADHD patients and healthy controls. The authors identified five key three-node motifs that characterize the underlying information flow. To characterize the function of nodes in motifs, they created the idea of node roles. When ADHD patients were compared to controls, some brain regions, such as the hippocampus, amygdala, thalamus, and temporal/occipital cortex, had aberrant and altered node role patterns. This shows that ADHD suffers from abnormal information processing, particularly in the limbic and visual systems, which may underpin emotional and attentional impairments. The findings shed light on ADHD's impaired brain connectivity at the mesoscale motif level. Overall, I found the manuscript interesting; however, I have some concerns, which I have listed below, that need to be addressed.
Major comments:
1. My main concern is that often Granger prediction performs poorly in data with low temporal resolution such as fMRI. The hemodynamic is major confounding factor, I am wondering have author thought about deconvolving the fMRI with hemodynamic response before conducting Granger causality analysis.
2. To increase readability of your paper please use “functional connectivity” instead of “information interaction”. The term “functional connectivity” has been established as a standard term for describing the communication between different brain regions. By using this standardized terminology, it becomes easier for researchers to understand and interpret the findings of studies related to brain connectivity.
3. Please clarify if you use Wiener–Granger causality in the text?
4. Under the participants section, please also introduce the NC dataset. It is not clear that whether the NC is part of the University of California Los Angeles (UCLA) Consortium for Neuropsychiatric Phenomics study, neither from text nor from Tabel 1.
5. Line 167, page 5: This section is not clear to me! Did the author looked into subsample of HCP (30%). Why is that? Please revise the text, it is very hard to follow.
6. Among all the 13 motifs, authors found 5 reproducible motif types (i.e., 3, 6, 9 ,12 and 13), however roles have assigned to only 3 types (i.e., 4: PPR, 6: PPO, 9: DPP). How do authors justify exclusion of motif type 12 and 13 from role?
7. Line 246 on page 7: "frequency spectrum" This is a very ambiguous term for describing Figure 3A, and it may confuse readers because it is frequently associated with spectral analysis and frequency domain representations in the MEEG literature. To avoid ambiguity, I recommend instead using the terms "frequency distribution" or "count distribution," as these terms accurately convey the concept of a graphical representation showing the frequency or count of occurrences of different values or ranges of a variable.
8. Please report the z-statistics of motifs with ID = 4, 6, 9, 12, and 13 separately. Also, is it not clear how the ratios in Figure 3C are obtained?
9. Please also run the node role analysis for HCP and report the values for the nodes shown in Figure 4.
10. Is there a main effect of group (ADHD, NC) for correlations between network measures and ASRS scores?
11. How much do authors think their findings:
a. change if they use dynamic measures of functional connectivity, for example see following:
· Kaboodvand N, Iravani B, Fransson P. Dynamic synergetic configurations of resting-state networks in ADHD. Neuroimage 2020;207:116347. https://doi.org/10.1016/j.neuroimage.2019.116347.
b. considering that ADHD is not homogeneous disorder and has a few subtypes, for example see following:
· Petrovic P, Castellanos FX. Top-Down Dysregulation-From ADHD to Emotional Instability. Front Behav Neurosci 2016;10:70. https://doi.org/10.3389/fnbeh.2016.00070.
· Iravani B, Arshamian A, Fransson P, Kaboodvand N. Whole-brain modelling of resting state fMRI differentiates ADHD subtypes and facilitates stratified neuro-stimulation therapy. Neuroimage 2021;231:117844. https://doi.org/10.1016/j.neuroimage.2021.117844.
Figures
Figure1B: More explanation is required in the caption about this panel, especially a brief description regarding how the motifs were identified.
Figure1C: same here, more explanation is required in the caption about this panel. Please summarize what the panel is trying to convey to readers.
Figure3A: It is unclear what the two bars (green and dark green) represent. Please include a figure legend and explain what these two colors represent in the figure caption.
Figure4D: I find it informative to also indicate the node role (PPR, PPO, PPD) in this panel. Node labels should be written out in the caption.
Minor comments:
1. Please write out the complete form of “NC” in its first appearance: page2, line 51
2. Please define and write out GCA at its first appearance.
3. Throughout the manuscript, the formulas have low resolution. Please revise the formulas and amend the resolution.
Minor language editing is required.
Author Response
Response to Reviewer 1 Comments
Point 1: My main concern is that often Granger prediction performs poorly in data with low temporal resolution such as fMRI. The hemodynamic is major confounding factor, I am wondering have author thought about deconvolving the fMRI with hemodynamic response before conducting Granger causality analysis.
Response 1: Thanks greatly for your helpful comment. I did not consider deconvolution of fMRI with hemodynamic response prior to Granger causality analysis. In this study, data preprocessing of the two datasets was conducted using the DPABI toolbox (Data Processing & Analysis for Brain Imaging, http://rfmri.org/dpabi), which is the current standard preprocessing pipeline. For the use of GCA method, we also refer to a lot of literature, such as Huang X et al., Azarmi F et al., Shi Y et al. In addition, when using the GCA method, I also consider the selection of model order p. For the multivariate autoregressive model order p, values too small can lead to a poor representation of the data, whereas values too large can lead to problems of model estimation. Therefore, we used the Akaike information criterion (AIC) and Bayesian information criterion (BIC) to select the optimal model order p. The results showed that, regardless of whether AIC or BIC was used, the error noise was the smallest when the model order p was set to 1. For the deconvolution method for fMRI, it could improve the availability of temporal information, and we would take this method into account in future studies. Thanks again!
References
Huang X, Zhang D, Wang P, et al. Altered amygdala effective connectivity in migraine without aura: evidence from resting‐state fMRI with Granger causality analysis[J]. The Journal of Headache and Pain, 2021, 22(1): 1-8.
Azarmi F, Ashtiani S N M, Shalbaf A, et al. Granger causality analysis in combination with directed network measures for classification of MS patients and healthy controls using task-related fMRI[J]. Computers in biology and medicine, 2019, 115: 103495.
Shi Y, Liu W, Liu R, et al. Investigation of the emotional network in depression after stroke: a study of multivariate Granger causality analysis of fMRI data[J]. Journal of Affective Disorders, 2019, 249: 35-44.
Point 2: To increase readability of your paper please use “functional connectivity” instead of “information interaction”. The term “functional connectivity” has been established as a standard term for describing the communication between different brain regions. By using this standardized terminology, it becomes easier for researchers to understand and interpret the findings of studies related to brain connectivity.
Response 2: Thanks greatly for your helpful suggestions. I apologize for the confusion about the term of “information interaction”. Throughout the article, we conducted a study based on the network motifs, which reflects the underlying information of the brain functional network. Then, we introduced the concept of node roles based on the similarity of ties in order to better characterize the node functions participating in network motifs. In other words, different node roles represent different ways in which information interacts. Here, we want to express the process of receiving and transmitting information between brain regions. We presume the term “functional connectivity” is not accurate enough to describe the dynamic process of information interaction. Thanks again!
Point 3: Please clarify if you use Wiener–Granger causality in the text?
Response 3: Thanks greatly for your helpful comment. I confirm that Wiener-Granger causality was used in the text. This article has been modified accordingly. . Thanks again!
To construct directed functional brain networks using resting-state fMRI data, we used the Wiener-Granger causality analysis (GCA) approach to estimate the mean time series of any pair of ROIs by computing the causation, which is a common method to study the causal relationship between variables on the base of causality of time series.
Point 4: Under the participants section, please also introduce the NC dataset. It is not clear that whether the NC is part of the University of California Los Angeles (UCLA) Consortium for Neuropsychiatric Phenomics study, neither from text nor from Tabel 1.
Response 4: Thanks greatly for your helpful suggestions. This article has been modified accordingly. . Thanks again!
The second dataset consisted of 50 rs-fMRI scans from normal controls and 42 rs-fMRI scans from individuals with ADHD. It was compared to the motif detection results of HCP datasets in order to establish a foundation for further research. They were provided by the Consortium for Neuropsychiatric Phenomics study at the University of California, Los Angeles (UCLA). Details of the dataset can be acquired from the OpenfMRI data-sharing webpage (https://www.openfmri.org/).
Point 5: Line 167, page 5: This section is not clear to me! Did the author looked into subsample of HCP (30%). Why is that? Please revise the text, it is very hard to follow.
Response 5: Thanks greatly for your helpful comment. I apologize for the distress caused by the inaccuracy of the language. In this study, we identified motif patterns in all networks of ADHD groups (threshold ranging from 0.1 to 0.4 with a partition interval of 0.05) and found that when the threshold was 30%, the population proportion of all kinds of network motifs in the two groups reached the maximum and remained stable. In order to maintain the same number of edges in two datasets, the coefficient matrix of the HCP directed brain functional networks was binaries by setting 30% thresholds. This article has been modified accordingly. Thanks again!
In this study, the coefficient matrix of the HCP directed brain functional networks was converted into binary form by applying a 30% threshold to ensure that the number of edges in the brain network matched that of the ADHD dataset.
Point 6: Among all the 13 motifs, authors found 5 reproducible motif types (i.e., 4, 6, 9 ,12 and 13), however roles have assigned to only 3 types (i.e., 4: PPR, 6: PPO, 9: DPP). How do authors justify exclusion of motif type 12 and 13 from role?
Response 6: Thanks greatly for your helpful comment. In order to better characterize the node functions participating in network motifs, we introduced the concept of node roles based on the similarity of ties between participating nodes. For example, motif13 contains three DPP role, and motif 12 contains one DPP role, one PPR role and one PPO role. We calculated each node roles of 5 reproducible motif types in the experimental part and explained them in Line 208 on page 6. Thanks again!
Point 7: Line 246 on page 7: "frequency spectrum" This is a very ambiguous term for describing Figure 3A, and it may confuse readers because it is frequently associated with spectral analysis and frequency domain representations in the MEEG literature. To avoid ambiguity, I recommend instead using the terms "frequency distribution" or "count distribution," as these terms accurately convey the concept of a graphical representation showing the frequency or count of occurrences of different values or ranges of a variable.
Response 7: I am grateful for your suggestions. I apologize for the confusion about the term of "frequency spectrum". Throughout the article, I've replaced "frequency spectrum" with "frequency distribution". Thanks again!
Point 8: Please report the z-statistics of motifs with ID = 4, 6, 9, 12, and 13 separately. Also, is it not clear how the ratios in Figure 3C are obtained?
Response 8: Thanks greatly for your helpful comment. The z-statistics of motifs with ID = 4, 6, 9, 12, and 13 separately of two datasets as following. The ratios in Figure 3C were obtained by dividing the number of subjects with a Z-score greater than 1.96 by the total number of people. Thanks again!
In this study, we examined resting-state functional MR images (fMRI) using two datasets to verify the consistency of the network motifs in the human brain. The first dataset was the HCP retest dataset. They could be identified twice to reflect the consistent motif patterns of the human brain and avoid randomness.
The z-statistics of motifs of the HCP retest dataset. (HCP-first test data)
The z-statistics of motifs of the HCP retest dataset. (HCP-second test data)
The z-statistics of motifs of the NC dataset.
The z-statistics of motifs of the ADHD dataset.
Point 9: Please also run the node role analysis for HCP and report the values for the nodes shown in Figure 4.
Response 9: Thanks greatly for your helpful comment. I have a little confusion about running the node role analysis for HCP. We examined resting-state functional MR images (fMRI) using two datasets for verifying the consistency of the network motifs in the human brain to demonstrate that these five types of network motifs are specific interaction patterns in functional directed networks of the human brain. It was a basis for further research of ADHD. Figure 4 shows the statistical difference analysis of ADHD role-degree. The node role analysis for HCP is not the research content of this paper. In addition, there was no control group for HCP data, so statistical test could not be performed. Finally, the values of role-degree for the nodes shown in Figure 4 as following Table 1 and has been placed in the supplementary material. Notably, reviewers 3 require a nonparametric test must be performed. We have employed the Wilcox test to assess group differences. This article has been modified accordingly. Thanks again!
Table 1 Brain regions showing significant differences based on node roles.
|
Role |
ROI |
Name |
Network |
p(FDR) |
ADHD(SD) |
NC(SD) |
|
DPP |
1 |
Precentral_L |
DMN |
0.003 |
138.35(172.63) |
260.12(212.59) |
|
DPP |
38 |
Hippocampus_R |
Subcortical |
0.007 |
197.28(218.65) |
117.04(158.30) |
|
DPP |
42 |
Amygdala_R |
Subcortical |
0.012 |
214.90(291.60) |
101.56(138.46) |
|
DPP |
67 |
Precuneus_L |
DMN |
0.03 |
236.78(225.62) |
436.96(448.27) |
|
PPR |
13 |
Frontal_Inf_Tri_L |
Attention |
0.014 |
150.66(148.59) |
76.66(67.66) |
|
PPR |
48 |
Lingual_R |
Visual |
0.045 |
107.40(73.47) |
187.12(167.18) |
|
PPR |
72 |
Caudate_R |
Subcortical |
0.04 |
117.33(114.4) |
213.56(238.34) |
|
PPR |
89 |
Temporal_Inf_L |
Attention |
0.033 |
117.71(103.07) |
173.64(135.84) |
|
PPO |
17 |
Rolandic_Oper_L |
Sensorimotor |
0.045 |
113.52(88.51) |
165.82(142.18) |
|
PPO |
79 |
Heschl_L |
Sensorimotor |
0.045 |
118.61(92.04) |
176.92(137.08) |
Point 10: Is there a main effect of group (ADHD, NC) for correlations between network measures and ASRS scores?
Response 10: Thanks greatly for your helpful comment. In this study, I have conducted Spearman correlations for group (ADHD, NC). For correlations within a single group, only one brain region was found to be present, which is weakly correlated and not very significant. The results associated with the group (ADHD, NC) are contained within the main text has been placed in the supplementary material. It was determined that there is a main effect of group on the correlations between network measures and ASRS scores. The Spearman correlations for single group are shown in Table 2. Thanks again!
Table 2 Brain regions showing significant correlations with ASRS scores for each group.
|
DPP role-degree |
PPR role-degree |
||||||||||
|
NC |
ADHD |
NC |
ADHD |
||||||||
|
ID |
r |
P |
ID |
r |
P |
ID |
r |
P |
ID |
r |
P |
|
38 |
0.187 |
0.204 |
38 |
0.157 |
0.332 |
13 |
0.05 |
0.735 |
13 |
0.193 |
0.233 |
|
42 |
0.165 |
0.263 |
42 |
0.275 |
0.043 |
48 |
-0.044 |
0.768 |
48 |
-0.079 |
0.627 |
Point 11: How much do authors think their findings:
- change if they use dynamic measures of functional connectivity, for example see following:
Kaboodvand N, Iravani B, Fransson P. Dynamic synergetic configurations of resting-state networks in ADHD. Neuroimage 2020;207:116347.https://doi.org/10.1016/j.neuroimage.2019.116347.
Response 11-a: Thanks greatly for your helpful comment. Kaboodvand et al. proposed a dynamical systems perspective to assess how the DMN over time recruits different configurations of network segregation and integration. From resting-state fMRI data, they extracted three different stable configurations of FC patterns for the DMN, namely synergies. Finally, they provided evidence supporting our hypothesis that ADHD differs compared to controls, both in terms of recruitment rate and topology of specific synergies between resting-state networks. I think this is very beneficial work and provides a new perspective for us to understand the pathological mechanism of ADHD. The question of adding the time dimension to network methods is also a topic we are working on. We have constructed a time-varying dynamic network about re-fMRI. We research the changes in node roles over time, as well as how the functional connectivity between normal and patient brain regions changes. There are already some preliminary results. Thanks again!
- considering that ADHD is not homogeneous disorder and has a few subtypes, for example see following:
- Petrovic P, Castellanos FX. Top-Down Dysregulation-From ADHD to Emotional Instability. Front Behav Neurosci 2016;10:70. https://doi.org/10.3389/fnbeh.2016.00070.
- Iravani B, Arshamian A, Fransson P, Kaboodvand N. Whole-brain modelling of resting state fMRI differentiates ADHD subtypes and facilitates stratified neuro-stimulation therapy. Neuroimage 2021;231:117844. https://doi.org/10.1016/j.neuroimage.2021.117844.
Response 11-b: Thanks greatly for your helpful comment. Generally, ADHD is typically classified into three categories - inattentive, hyperactive, and combined. I believe this division is advantageous for treating ADHD. According to the type of ADHD patients, the corresponding damaged brain areas will be different. In this study, the dataset did not refer to ADHD subtypes. Our findings come from examining all people with ADHD, and there are certainly some more interesting results when we drill down to the specific subtypes of ADHD. Thanks again!
Point 12: Figure1B: More explanation is required in the caption about this panel, especially a brief description regarding how the motifs were identified.
Response 12: I am grateful for your suggestions. I apologize for the confusion about this panel. According to the suggestion of reviewer 3, the Fig. 1 has been moved to the end of the Materials and Methods Section with more explanation. This article has been modified accordingly. . Thanks again!
(B) Identified network motifs of human brain effective networks in two datasets. Through the calculation of formula (6), (7) and (8) it was found that the specific network motifs of human brain effective network are 4,6,9,12 and 13.
Point 13: Figure1C: same here, more explanation is required in the caption about this panel. Please summarize what the panel is trying to convey to readers.
Response 13: I am grateful for your suggestions. I apologize for the confusion about this panel. According to the suggestion of reviewer 3, the Fig. 1 has been moved to the end of the Materials and Methods Section with more explanation. This article has been modified accordingly. . Thanks again!
(C) Analysis motif-based node roles in ADHD dataset. According to the results of the identifiable motifs, we categorized the node roles into three types: DPP, PPO, and PPR. Through the calculation of formula (9), (11) and (12), we used statistical analysis to explore the differential brain regions, discrepant effective connectivity and changing node roles.
Point 14: Figure3A: It is unclear what the two bars (green and dark green) represent. Please include a figure legend and explain what these two colors represent in the figure caption.
Response 14: I am grateful for your suggestions. I apologize for the confusion about the two bars (green and dark green). This article has been modified accordingly. Thanks again!
In the HCP retest dataset, the green bar represents the first test data for HCP, while the dark green bar represents the second test data for HCP. In the ADHD dataset, the dark green bar represents the data for normal controls, while the dark red bar represents the data for individuals with ADHD.
Point 15: Figure4D: I find it informative to also indicate the node role (PPR, PPO, PPD) in this panel. Node labels should be written out in the caption.
Response 15: I am grateful for your suggestions. I have written out the node labels in Line 299 on page 9. This article has been modified accordingly. Thanks again!
The PreCG.L, the HIP.R, the AMYG.R and the PCUN.L belong to DPP role; the ROL.L and the HES.L belong to PPO role; the IFGtriang.L, the LING.R, the CAU.R and the ITG.L belong to PPR role.
Point 16: Please write out the complete form of “NC” in its first appearance: page2, line 51
Response 16: Thanks greatly for your helpful comment. I have written out the complete form of “NC” in Line 47 on page 2. Thanks again!
Point 17: Please define and write out GCA at its first appearance.
Response 17: Thanks greatly for your helpful suggestions. I have defined and written out GCA at its first appearance. This article has been modified accordingly. . Thanks again!
To construct directed functional brain networks using resting-state fMRI data, we used the Wiener-Granger causality analysis (GCA) approach to estimate the mean time series of any pair of ROIs by computing the causation, which is a common method to study the causal relationship between variables on the base of causality of time series.
Point 18: Throughout the manuscript, the formulas have low resolution. Please revise the formulas and amend the resolution.
Response 18: Thanks greatly for your helpful comment. The equations have been rearranged to make sure all equations clear. Thanks again!

Reviewer 2 Report
The present study brings a new perspective through the application of the network motif approach to control and ADHD’s fMRI analysis during a resting state. The idea behind this is to add information and comprehension on meso-scale topology information of brain function.
The authors decide to set 3 as the number of node-motifs. These meso-scale networks are considered relevant and recurring connectivity patterns and with them, it is possible to determine the information flow. They focus on ADHD versus control’s differences between effective connectivity and on different brain regions' involvement.
The more obvious limitation is explained in the limitation section (the ad-hoc selection of 3-node motifs). The manuscript is interesting and clearly written. I have some suggestions/ comments:
- Authors should explain in more detail in the discussion what their approach differentiates from classical effective connectivity analysis on sources, or what additional information they provide. When reading the discussion, one wonders if they really bring more details to the existing literature. I think that it is the case but it is not clearly explained.
- Also in the discussion section, authors could open the perspective of using their approach on active tasks as typical Go-No Go paradigms. This could be a kind of validation (even if further information will be gained) as the brain areas/nodes have been largely studied (Please consider Zarka et al., 2020 and 2021 where taking advantage of the good temporal resolution of EEG, authors showed that the areas contributing to ERP in ADHD and TDC differ from the early steps of visual-attentional processing and reveal an overinvestment of the executive networks interfering with the activity of the dorsal attention network in children with ADHD. Could the temporal dimension be added (like on ERP from EEG) to the network approach? That would open a lot of exciting possibilities.
- In Zarka et al 2021, authors observed the reduced contribution of the dorsolateral prefrontal cortex, the insula, and the cerebellum. What about the cerebellum? Do the Authors have any information on the nodes in the cerebellum?
Author Response
Response to Reviewer 2 Comments
Point 1: Authors should explain in more detail in the discussion what their approach differentiates from classical effective connectivity analysis on sources, or what additional information they provide. When reading the discussion, one wonders if they really bring more details to the existing literature. I think that it is the case but it is not clearly explained.
Response 1: Thank you very much for your suggestions. I apologize for the unclear about detail in the discussion where the proposed approach differentiates from classical effective connectivity analysis. I have updated the discussion section and supplemented it with further discussion. Thanks again!
Point 2: Also in the discussion section, authors could open the perspective of using their approach on active tasks as typical Go-No Go paradigms. This could be a kind of validation (even if further information will be gained) as the brain areas/nodes have been largely studied (Please consider Zarka et al., 2020 and 2021 where taking advantage of the good temporal resolution of EEG, authors showed that the areas contributing to ERP in ADHD and TDC differ from the early steps of visual-attentional processing and reveal an overinvestment of the executive networks interfering with the activity of the dorsal attention network in children with ADHD. Could the temporal dimension be added (like on ERP from EEG) to the network approach? That would open a lot of exciting possibilities.
Response 2: Thank you very much for your suggestions. You provided a good idea for me. Applying the proposed approach to active tasks as a typical Go-no-Go paradigm will be our next step. I'll talk about that in the discussion section as well. The question of adding the time dimension (like on ERP from EEG) to network methods is also a topic we are working on. We have constructed a time-varying dynamic network about re-fMRI. We research the changes in node roles over time, as well as how the interaction of information between normal and patient brain regions changes. There are already some preliminary results. Thanks again!
Point 3: In Zarka et al 2021, authors observed the reduced contribution of the dorsolateral prefrontal cortex, the insula, and the cerebellum. What about the cerebellum? Do the Authors have any information on the nodes in the cerebellum?
Response 3: Thanks greatly for your helpful comment. In this study, we excluded cerebellar information for comparison with the previous motif recognition literature (consistency of 90 brain map templates). If adding information about the cerebellum, I think there will be some interesting findings as well. I will add this question to the "limitations and future directions" section and complete the cerebellar information in further research. Thanks again!

Reviewer 3 Report
The study explores abnormal and changing information interactions based on network motifs, providing important evidence for understanding information interactions at the mesoscale level in ADHD patients.
The paper is well written, the topic interesting, and the methodology innovative.
However, there are some small issues that should be fixed. In particular:
1) The equations are almost unreadable due to the current formatting.
2) The use of the T-test requires verification of the normality of the distributions of the data on which one operates. In the absence of this verification, a nonparametric test must be performed (see, Montgomery, Douglas C., and George C. Runger. Applied statistics and probabilities for engineers. John Wiley & Sons, 2010.)
3) Fig. 1: move to the end of the Materials and Methods Section otherwise many of its labels are not properly introduced
4) Line 234: Move the web link into the reference
Author Response
Response to Reviewer 3 Comments
Point 1: The equations are almost unreadable due to the current formatting.
Response 1:Thanks greatly for your helpful comment. The equations have been rearranged to make sure all equations clear. Thanks again!
Point 2: The use of the T-test requires verification of the normality of the distributions of the data on which one operates. In the absence of this verification, a nonparametric test must be performed (see, Montgomery, Douglas C., and George C. Runger. Applied statistics and probabilities for engineers. John Wiley & Sons, 2010.)
Response 2: I am grateful for your comment. We have employed the Wilcox test to assess group differences. This article has been modified accordingly. Thanks again!
Point 3: Fig. 1: move to the end of the Materials and Methods Section otherwise many of its labels are not properly introduced.
Response 3: I am grateful for your suggestions. The Fig. 1 has been moved to the end of the Materials and Methods Section with more explanation.
Point 4:Line 234: Move the web link into the reference.
Response 4: Thank you very much for your suggestions. I have moved the web link into the reference. Thanks again!

Reviewer 4 Report
1. Abstract was too long.
2. Keywords given in the manuscript was not sufficient.
3. Need more explanation for resting-state functional MR images (fMRI) using two datasets.
4.Any specific contribution made from the study for society
5.More Explanation needed for data acquisition and pre-processing.
6. Any specific protocols followed by the authors for data acquisition.
7. Equations given in the 2.3, 2.4 and were was not clear.
8. Result inferences given by the authors were too long. SO i kindly request the author to minimize the result content.
Author Response
Response to Reviewer 4 Comments
Point 1:. Abstract was too long.
Response 1: Thank you very much for your suggestions. The abstract contents have been minimized. Thanks again!
Network motif analysis approaches provide insights into the complexity of the brain's functional network. In recent years, attention-deficit/hyperactivity disorder (ADHD) has been reported to result in abnormal information interactions on macro- and micro-scale functional networks. However, most existing studies remain limited due to potentially ignoring meso-scale topology information. To address this gap, we aimed to investigate functional motif patterns in ADHD to unravel the underlying information flow and analyze motif-based node roles to characterize the different information interaction ways for identifying the abnormal and changing lesion sites of ADHD. The results showed that the interaction functions of the right hippocampus and the right amygdala were significantly increased, which could lead patients to develop mood disorders. The information interaction of the bilateral thalamus changed, influencing and modifying behavioral results. Notably, the capability of receiving information in the left inferior temporal and the right lingual gyrus decreased, which may cause patients difficulty in processing visual information in a timely manner, resulting in inattention. This study revealed abnormal and changing information interactions based on network motifs, providing important evidence for understanding information interactions at the meso-scale level in ADHD patients.
Point 2:Keywords given in the manuscript was not sufficient.
Response 2: Thank you very much for your suggestions. The new keywords have been added according to the research content. Thanks again!
Keywords: ADHD; brain effective network; network motifs; information interaction; node roles; abnormal interaction; changing roles
Point 3:Need more explanation for resting-state functional MR images (fMRI) using two datasets.
Response 3: Thanks greatly for your helpful suggestions. I apologize for the distress caused by the partial deletion of the dataset introduction due to its previous length. The text will be modified accordingly. Thanks again!
We examined resting-state functional MR images (fMRI) using two datasets to verify the consistency of the network motifs in the human brain. (1) The first dataset was the HCP retest dataset, which contains 45 resting-state functional MRI (rs-fMRI) scans from the Human Connectome Project (HCP) Retest data release [32]. They could be identified twice to reflect the consistent motif patterns of the human brain and avoid randomness. Details of the dataset can be found on the HCP website (http://www.humanconnectome.org/). (2) The second dataset consisted of 50 rs-fMRI scans from normal controls and 42 rs-fMRI scans from individuals with ADHD. It was compared to the motif detection results of HCP datasets in order to establish a foundation for further research. They were provided by the Consortium for Neuropsychiatric Phenomics study at the University of California, Los Angeles (UCLA). De-tails of the dataset can be acquired from the OpenfMRI data-sharing webpage (https://www.openfmri.org/).
Point 4:Any specific contribution made from the study for society.
Response4: Thanks greatly for your helpful comment. In recent years, attention-deficit/hyperactivity disorder (ADHD) has been reported to result in abnormal information interactions on macro-scale or micro-scale functional networks. However, most existing studies remain limited by potentially ignoring the meso-scale topology information. To address this gap, we investigated 3-node functional motif patterns in ADHD to unravel the underlying information flow and analyze motif-based node roles to characterize the different information interaction ways for identifying the abnormal and changing lesion sites of ADHD. This study revealed abnormal and changing information interactions based on network motifs, providing important evidence for understanding information interactions at the meso-scale level in ADHD patients. Thanks again!
Point5:More Explanation needed for data acquisition and pre-processing.
Response 5: Thanks greatly for your helpful comment. I apologize for the confusion about the data acquisition and pre-processing due to its previous length. The text will be modified accordingly. Thanks again!
The HCP imaging data were acquired on a customized 3T Siemens connectome-Skyra 3T scanner using a multiband sequence. Each participant completed two rs-fMRI scanning sessions 140 days apart. The parameters were as follows: repetition time (TR) = 720 ms, echo time (TE) = 33.1ms, slice thickness = 2 mm, slices = 72, flip angle = 52◦, and duration = 14 min and 33 s (1200 TRs). The magnetic resonance images of ADHD patients were acquired using a 3 T Siemens Trio scanner. Each participant completed one rs-fMRI scanning session. The parameters were as follows: repetition time (TR) = 2 s, echo time (TE) = 30ms, slice thickness = 4 mm, slices = 34, flip angle = 90◦ field of view (FOV) = 192 mm, matrix = 64 × 64, and duration = 5 min and 4 s (152 TRs). During the entire scanning, the participants of the two datasets were asked to relax and close their eyes, but not fall asleep.
Data preprocessing of the two datasets was conducted using the DPABI toolbox (Data Processing & Analysis for Brain Imaging, http://rfmri.org/dpabi) [33]. Firstly, we discarded the first ten volumes of the signal considering the adaptability of the environment and corrected the first slice timing and head motion of the remaining data. Subsequently, the data generated were normalized relative to the Montreal Neurological Institute (MNI) standard space in order to compensate for individual brain variations. Next, to reduce the inexactness of registration and enhance the signal-to-noise ratio, spatial smoothing was conducted using a Gaussian kernel with 6 mm full-width at half-maximum (FWHM). Additionally, bandpass filtering (0.01≤f≤0.1 Hz) was implemented on the image to trans-form the time series into the frequency domain and calculate the energy in the lower frequency band. Finally, resting-state scans were parcellated into 90 regions of interest (ROIs) using the automatic anatomical marker template (AAL atlas) [34] and the time series were extracted. Notably, the cerebellum was excluded from this study.
Point 6:Any specific protocols followed by the authors for data acquisition.
Response 6: Thanks greatly for your helpful comment. Since both two datasets are public datasets, we could download them from the website as long as we have registered the account information. There is no any specific protocol followed. We have indicated the data sources in the "Data Availability Statement" section. Thanks again!
Point 7:Equations given in the 2.3, 2.4 and 2.5 were was not clear.
Response 7: Thanks greatly for your helpful suggestions. The equations have been rearranged to make sure all equations clear. Thanks again!
Point 8:Result inferences given by the authors were too long. So I kindly request the author to minimize the result content.
Response 8: Thanks greatly for your helpful suggestions. The result and discussion contents have been minimized. Thanks again!

Round 2
Reviewer 1 Report
I thank the authors for their effort for amending their manuscript by incorporating my comments. However, there are some unclear points remained that I would like to mention. I believe some of the points, that are discussed in the cover letter, are worth to mention in the manuscript:
1. It is important to point out that in the manuscript, that your approach might perform better if the fMRI signals are first deconvolved with the hemodynamic response.
a. Also, it is important to mention in the manuscript that how you estimated the model order p. It would be helpful if you report the AIC and BIC in the manuscript or a supplementary figure where you show the values for AIC and BIC as function of different tested p. Please clarify for how many values of p did you test for?
2. Per explanation of the authors, now I understand that “function connectivity” is not an accurate term. However, the authors need to explain up front in the manuscript their definition of the “information exchange”.
3. Please write in the manuscript the future directions as were discussed in the revision and cover letter:
a. The use of time varying measure, and how it might be beneficial and complement the previous research in this domain.
b. How can one use your method to identify subgroups of ADHD and who it can help clinical diagnosis.
i. Also please discuss the use of your method to define new covert subtypes and discuss with regard to the references that I provided in the first round of revision.
Minor comments:
1. Please keep using the acronyms after their introduction, for example on page 2, lines 82. Authors wrote “The second dataset consisted of 50 rs-fMRI scans from normal controls and 83 42 rs-fMRI scans…”; the term “normal controls” is already introduced as NC, and therefore please keep referring to it as NC. Please revise manuscript for similar potential problems.
a. NC is defined at multiple occasions:
i. Page 2, line 47
ii. Page2, line 57
Please make sure the acronyms are defined only at the first appearance of the term and kept being used in the rest of the text.
2. Threshold sometime are mentioned as percentage (page 5, line 176: 30%) and sometimes as decimals (page 5, line 181: 0.1 to 0.4). It is important to keep similar notation.
a. Please also clarify in the manuscript if 30% threshold referring to keeping 30% of the strongest connection.
3. Figure 7, Please briefly interpret what each panel’s result mean in the legend. The figures together with their legend should to some extend self-explanatory and independent from the text.
Minor editing is required to increase the readability of the manuscript.
Author Response
Dear reviewer, thanks greatly for your helpful comment. Since the cover letter contains formulas and pictures, this cannot be shown completely. Please see the attachment. Thanks again!
